# OpenReview forum: "Cowpox: Towards the Immunity of VLM-based Multi-Agent Systems"
_ICML.cc/2025/Conference — ICML 2025 poster_

### Official Review · Reviewer_iPpj · 2025-02-19

**Overall Recommendation:** 3

**Summary:**

This paper introduces COWPOX, a novel defense approach designed to enhance the robustness of multi-agent systems (MAS) against adversarial attacks. Vision-Language Model (VLM)-based agents, which perceive and interact with their environment through vision and language, are integral to MAS. However, existing MAS designs often overlook robustness, allowing exploits to spread and compromise system integrity. COWPOX addresses this by implementing a distributed mechanism that limits the expected number of infections and improves agent recovery rates. The core innovation is the generation and distribution of a special cure sample that immunizes agents pre-exposure and aids in the recovery of infected ones. The paper empirically demonstrates COWPOX's effectiveness and provides theoretical robustness guarantees.

## update after rebuttal

I have reviewed the authors' response. They mentioned not over-relying on llm-judge and supplemented the experiments with a new agent architecture, which addressed my concerns. Therefore, I maintain my positive score.

**Claims And Evidence:**

Well supported.

**Essential References Not Discussed:**

Not at all, to my knowledge, AgentSmith is indeed the first article to address the attack (ICML 2024), and I believe this is a defense article that closely follows the SOTA.

**Experimental Designs Or Analyses:**

Appears to be a comprehensive experiment. No further questions.

**Methods And Evaluation Criteria:**

Make sense.

**Other Comments Or Suggestions:**

N/A

**Other Strengths And Weaknesses:**

Overall, I believe this article is well done. From a methodological design perspective, I find it reasonable, and the presentation of images and formulas appears to be meticulously polished. However, I am not familiar with RAG and propagation-based jailbreaking, so I am unable to make a judgment on the technical details of the article. I would prefer to leave the assessment of the article to other reviewers and the Area Chair AC.

Weakness:
1. The design of multi-agent systems is still in the exploratory phase, which may give rise to multiple schools of thought, yet the authors have only discussed one. In other words, the generalizability of the proposed defense mechanism may be questionable.

**Questions For Authors:**

1.Does cowpox reduce the performance of the system when all agents are clean? In other words, will the false positive examples from Output Analysis Module affect the system? I am quite curious about some specific examples of the-LLM-as-a-inspector.

**Relation To Broader Scientific Literature:**

N/A

**Theoretical Claims:**

No

---

> ### Author Rebuttal · Authors · 2025-03-31
>
> We'd like to appreciate your praise on the completeness of the paper and the design of our method. Hope our response can solve your concerns.
>
> > ***1. About the LLM-based inspector***
>
> We agree that the inspector is important to our method. We give the details of the performance of the inspector used in the paper below:
> ||ACC(%)|FPR(%)|FNR(%)|
> |-|-|-|-|
> |1 try|84.7|2.8|12.5|
> |3 tries|89.1|7.9|3.0|
> - Settings: The evaluation is conducted on a combination of malicious outputs from Advbench and normal(benign) outputs from the ordinary chat history of our agents. We test the inspector in 1-turn and 3-turn settings. For the multi-turn test, a sample is labeled as harmful if it is classified as harmful in **ANY** turn.
> - Analysis: Overall, although not able to achieve 100% accurate detection, **our inspector is effective enough for Cowpox**. This is because:
>     - **Only very few benign samples will be misclassified.** Cowpox agents account for only a very small portion of the total, so less than 2% of benign samples will be misclassified.
>     - **Misclassifying benign samples has little impact on the system.** The cure crafted based on misclassified benign samples still perfectly contains its original information. So there will be almost no effects.
>     - **FNR approaches 0 when the test round increases.** The system only needs the virus to be detected in any chat round once to achieve effective defense. Cowpox does not require a very strong inspector.
>
> > ***2. The generalizability of the proposed defense mechanism may be questionable.***
>
> **The generalizability can be demonstrated by the following two experiments.** We modify the original system to simulate diverse systems.
>
> - **Effective in the system of different structured communication settings.**
>     - Settings: We assume that the system is hierarchical. The agents are divided into 8 groups, and can only do the pair-wise chat with their group members. In each group, there are 3 manager agents, which communicate with manager agents of other groups every 4 rounds. Such hierarchical structures are often adopted in works like AutoGen [[1]](https://arxiv.org/abs/2308.08155).
>     - Results: [link](https://imgur.com/2gN812H).
>     - Analysis: We can see that the virus infects agents more slowly, and so does the Cowpox cure the agent. This is because the hierarchical structure can temporarily isolate the infected agents.
>
> - **Effective across diverse underlying base models and RAG systems**
>     - Settings: We adopt 2 VLM models (LlaVA-1.5, InstructBLIP) and 2 RAG encoders (CLIP, DINO V2) in the experiment. The agent chooses its base model and RAG encoder *randomly* initially to form a multi-agent that consists of heterogeneous agents.
>     - Results: [link](https://imgur.com/LOQiZ0R).
>     - Analysis: From the figure, The virus in this system performs worse, while Cowpox is almost equally effective. This is because the cure only targets the RAG system, therefore fewer models are involved in crafting it, making the optimization easier.
>
>
> ---
>
> > ***Reference***
> 1. Wu, Qingyun, et al. "Autogen: Enabling next-gen llm applications via multi-agent conversation." arXiv preprint arXiv:2308.08155 (2023).

---

> > ### Comment · Reviewer_iPpj · 2025-04-02
> >
> > I have reviewed the authors' response. They mentioned not over-relying on llm-judge and supplemented the experiments with a new agent architecture, which addressed my concerns. Therefore, I maintain my score and recommend the paper.

---

> > > ### Author Response · Authors · 2025-04-03
> > >
> > > We respectfully appreciate your recommendation and your decision to maintain a positive attitude towards our work.

---

### Official Review · Reviewer_sDAE · 2025-03-10

**Overall Recommendation:** 2

**Summary:**

This paper targets the problem that there are attack agents in a vision-language-model-based multi-agent systems. The authors propose a defense method named Cowpox. It generate and distribute cure samples, which will be scored higher in the retrieval-augmented generation and can help those injected agents. Experiments show the effectiveness of the proposed method.

## update after rebuttal

My major concern is on the limited generality of the targeted attack scenarios and proposed defense method. While the authors tried to address this by showing more cases, it seems like the content during rebuttal somehow deviates from what the initial manuscript looks like.

I choose to maintain my current rating and suggest the authors to entirely rewrite the paper to make the attack and defense generalizable to diverse scenarios rather than targetting only one scenario throughout the paper.

**Claims And Evidence:**

yes

**Essential References Not Discussed:**

no

**Experimental Designs Or Analyses:**

yes

**Methods And Evaluation Criteria:**

yes

**Other Comments Or Suggestions:**

no

**Other Strengths And Weaknesses:**

## Strengths
1. The paper is in a good structure and is easy to follow
2. The effectiveness of the proposed method is supported by sufficient experimental results.

## Weaknesses
1. The foremost concern for me is motivation and the practicality of the targeting scenario. In practice, multi-agent systems are usually applied to solve complex tasks or to simulate real-world scientific experiments. In these application scenarios, one would control all of the agents to achieve the goals and there seems to be no chance that an attack agent will be involved, for example, one may uses open-sourced well-performed LLMs released by Meta or Qwen. And these models will not diliberately attack the system. That is, if there is no such attack in practice, it is unclear for me what such research (this paper) is necessary.

2. Meanwhile, the proposed defense method is too specific. It is designed on the system proposed by Gu et. al., 2024 and verified on this system. This makes the scope of this paper narrow.

**Questions For Authors:**

no

**Relation To Broader Scientific Literature:**

unclear

**Theoretical Claims:**

no

---

> ### Author Rebuttal · Authors · 2025-03-31
>
> Thank you for your valuable feedback, which we believe will improve the paper. We appreciate that you liked the structure of our paper and our method’s effectiveness. Please find our responses below.
> > ***1. The practicality of the targeting scenarios***
>
> We agree with your comments that currently, the user has full access to all the agents is a common setting in many well-known multi-agent systems. However, this setting does not apply to all of the multi-agent systems. Our scenario is practical, especially in the following scenarios.
> - **Multi-embodied-agent system.** Pioneering works like [Co-LLM-Agent](https://github.com/UMass-Embodied-AGI/CoELA) create systems with multiple embodied agents operating in the physical world. In this scenario, an attacker can introduce the attack agent by simply putting it somewhere close to the benign agents to infect the system.
> - **Multi-agent system on edge devices.** There are works [2-3] investigating distributive multi-agent systems. In these systems, the agents are implemented on different edge devices like mobile phones or vehicles, where it is hard for the user to access all of the agents, making it possible for a malicious agent to get involved.
> - **Blockchain-based decentralized multi-agent systems.** Systems like [SingularityNET](https://singularitynet.io), [Fetch.AI](https://www.fetch.ai), and [HyperCycle](https://www.hypercycle.ai) **allow the agents owned by different users to collaborate, share data, and learn from each other anonymously, which further realizes the attack surface.** An attack started by a malicious agent can be easily conducted in these scenarios, and distributive countermeasures like Cowpox seem to be the only solution.
> - Finally, **The threat of the infectious attack could come NOT from the agents, but from the working environment of the system.** Even in a situation where the user has full access, A virus in the environment can always turn an originally benign agent into an attack agent by infecting it in the wild. For example, multi-agent systems like [Camel](https://www.camel-ai.org) and [BabyAGI](https://github.com/yoheinakajima/babyagi) allow the agents in the system to access the Internet, where the virus sample might be accidentally obtained by the agent and infect it, turning this unlucky agent into an attacker. The attack can also be achieved by other approaches such as poisoning the RAG database of certain agents.
>
> > ***2. The proposed defense method is too specific***
>
> We fully understand your concerns about the generalizability of Cowpox. We'd like to claim that our method is a general defense strategy against infectious attacks.
>
> - **Our method is designed to deal with an attack paradigm instead of a specific attack.** The core idea of Cowpox is **to isolate the suspicious sample from the agent by preventing it from being retrieved.** This indicates that our method is able to interrupt the transmission process of the infectious attack in ANY multi-agent system with RAG.
> - **The alignment in the tested system helps us better evaluate our defense.** The attack performance of Gu et. al. is perfectly demonstrated in their paper. **We mainly test our defense on their system to fairly show its effectiveness.**
>
> To further demonstrate the generalizability, we investigate two different multi-agent systems to show that Cowpox can work in diverse settings:
>
> - **Effective in the system of different structured communication settings.**
>     - Settings: We assume that the system is hierarchical. The agents are divided into 8 groups, and can only do the pair-wise chat with their group members. In each group, there are 3 manager agents, which communicate with manager agents of other groups every 4 rounds. Such hierarchical structures are often adopted in works like AutoGen [[1]](https://arxiv.org/abs/2308.08155).
>     - Results: [link](https://imgur.com/2gN812H).
>     - Analysis: We can see that the virus infects agents more slowly, and so does the Cowpox cure the agent. This is because the hierarchical structure can temporarily isolate the infected agents.
>
> - **Effective across diverse underlying base models and RAG systems**
>     - Settings: We adopt 2 VLM models (LlaVA-1.5, InstructBLIP) and 2 RAG encoders (CLIP, DINO V2) in the experiment. The agent chooses its base model and RAG encoder *randomly* initially to form a multi-agent that consists of heterogeneous agents.
>     - Results: [link](https://imgur.com/LOQiZ0R).
>     - Analysis: From the figure, The virus in this system performs worse, while Cowpox is almost equally effective. This is because the cure only targets the RAG system, therefore fewer models are involved in crafting it, making the optimization easier.
>
> # References
> 2. Zhang, Chi, et al. "Appagent: Multimodal agents as smartphone users." arXiv preprint arXiv:2312.13771 (2023).
>
> 3. Wang, Junyang, et al. "Mobile-agent: Autonomous multi-modal mobile device agent with visual perception." arXiv preprint arXiv:2401.16158 (2024).

---

### Official Review · Reviewer_gaAP · 2025-03-12

**Overall Recommendation:** 3

**Summary:**

This paper addresses the vulnerability of Vision Language Model (VLM)-based multi-agent systems to infectious jailbreak attacks, where a compromised agent can spread malicious content to other agents, undermining the system's robustness. The paper proposes a novel defense mechanism called COWPOX, which aims to provide immunity to such systems.

**Claims And Evidence:**

Yes

**Essential References Not Discussed:**

No

**Experimental Designs Or Analyses:**

yes

**Methods And Evaluation Criteria:**

Yes

**Other Comments Or Suggestions:**

no

**Other Strengths And Weaknesses:**

Strengths

Novelty: The paper tackles a critical and relatively unexplored security vulnerability in VLM-based multi-agent systems.

Well-Defined Problem: Clearly articulates the problem of infectious jailbreak attacks and its implications for system robustness.

Comprehensive Approach: Combines a practical defense mechanism (COWPOX) with theoretical analysis and empirical validation.

Solid Empirical Results: The experimental section provides evidence for the effectiveness of COWPOX under different attack strategies and defender abilities.

Clear Presentation: The paper is generally well-written and organized, with clear explanations of the proposed method and experimental setup.

Addresses Practical Constraints: Acknowledges and addresses real-world constraints such as the limited control a defender might have over the agents in the system.

Weaknesses

Scalability Concerns: While the paper addresses the number of agents, further discussion on the computational overhead of the LLM-based inspection (Output Analysis Module) in large-scale systems would be beneficial. How does the complexity of this module scale with the number of agents or the size of the chat histories?

Adaptive Attack Complexity: While the paper mentions resistance to adaptive attacks, the complexity of the adaptive attack used in the evaluation could be further elaborated. Are there more sophisticated adaptive strategies that could potentially circumvent COWPOX?

Generalizability: The experiments are conducted with a specific VLM (Llava-1.5-7B) and a specific multi-agent system setup. It would be useful to discuss the potential generalizability of COWPOX to other VLM architectures and multi-agent system designs.

**Questions For Authors:**

see strength and weakness part.

**Relation To Broader Scientific Literature:**

The paper builds upon existing research on jailbreak attacks against VLMs , particularly infectious jailbreak attacks in multi-agent systems . It differentiates itself by proposing a defense mechanism tailored to multi-agent systems, addressing the limitations of individual model defenses.

**Theoretical Claims:**

Yes. The proposition 4.1 is checked.

---

> ### Author Rebuttal · Authors · 2025-03-31
>
> We appreciate that you liked the novelty of our paper and all the other strengths stated. Please find our responses below.
>
>
> > ***1. Scalability Concerns***
>
>
> **The computational overhead of the LLM-based inspector is linearly related to the chat rounds, and the numbers of the cowpox agent, and is not related to the length of the chat history**, as it only examines the output in the current chat round in our design. A more capable inspector that also goes through the chat history can be implemented though in case of a more sophisticated attack. We will discuss this issue in the revised paper.
>
>
> > ***2. Adaptive Attack Complexity***
>
>
> The adaptive attack discussed in the paper might seem simple, but it is exclusively designed for Cowpox. Below we rationale our claim:
>
>
> - **The fundamental mechanism of Cowpox.** The cure samples crafted by Cowpox eliminate the virus sample by gaining a higher RAG score. This means the virus cannot be retrieved from the database of the agent so as to interrupt the transmission.
>
>
> - **The adaptive attack targets the fundamental mechanism of Cowpox.** To target this mechanism, the adaptive attack tries to craft a sample that could score even higher than the cure sample. This is guaranteed by comparing the average RAG score of the cure sample and that of the virus, and stopping the optimization until the RAG score of the adaptive virus exceeds that of the cure sample.
>
>
> **We considered a more sophisticated version of the adaptive attack**, in conclusion, **Cowpox in this case is still effective.**
>
>
> - Settings: we adopted meta-learning to make the jailbreak target more resilient to the recovery process of Strategy ① discussed in the paper. The adaptive virus is crafted by a dual optimization process:
>     - The inner loop simulates the process of Strategy ①.
>     - The outer loop maximizes the RAG score and minimizes the loss between the VLM output and the target outputs.
>
>
> - Results: [link](https://imgur.com/PossSSf)
>
>
> - Analysis: We find that the so-crafted adaptive virus tends to be less effective, as demonstrated in the provided link. On the other hand, it takes more rounds for the cure sample to recover all the agents, indicating the effectiveness of the meta-learning.
>
>
> Lastly, it is always possible that more sophisticated, adaptive attacks will be developed in the future that circumvents our defense and would be interesting to study as a follow-up to our work.
>
> > ***3. Generalizability***
>
> Thank you for your valuable advice. We conduct experiments on agents based on different VLM and RAG systems.
> **The effectiveness holds when heterogeneous agents coexist in the same system**,
> - Settings: we adopt 2 VLM models (LlaVA-1.5, InstructBLIP) and 2 RAG encoders (CLIP, DINO V2) in the experiment. The agent chooses its base model and RAG encoder *randomly* initially to form a multi-agent that consists of heterogeneous agents.
> - Results: [link](https://imgur.com/LOQiZ0R).
> - Analysis: From the figure, The virus in this system performs worse, while Cowpox is almost equally effective. This is because the cure only targets the RAG system, therefore fewer models are involved in crafting it, making the optimization easier.

---

> > ### Comment · Reviewer_gaAP · 2025-04-05
> >
> > Thanks for author's response. I keep my rate unchanged.

---

> > > ### Author Response · Authors · 2025-04-05
> > >
> > > We respectfully appreciate your valuable comments and response.

---

### Official Review · Reviewer_VWXx · 2025-03-23

**Overall Recommendation:** 3

**Summary:**

The paper presents Cowpox, a method to prevent infectious jailbreaks in multi-agent systems of VLMs. A single agent can start out with an adversarial example that can affect other agents in the system, and Cowpox provides a method to override this adversarial example with a small number of agents part of the defense mechanism. This reduces the number of ‘infected’ agents in the system.

**Claims And Evidence:**

Most claims are justified but some claims around theoretical robustness guarantees depend on a large number of agents and rounds, stable RAG scores, and other assumptions. Depends heavily on the inspector prompt and doesn’t provide any analysis on false negative rates in this context.

**Essential References Not Discussed:**

N/A

**Experimental Designs Or Analyses:**

All agents use the same model and CLIP RAG setup, which does not reflect real world multi-agent systems. Moreover, a lot of reliance is placed on CLIP or BLEU scores for evaluating outputs, but this may not accurately represent system degradation. Moreover, experiments assume a random communication structure which limits real-world relevance significantly. The only adaptation tested is a single new virus, which is too simplistic for the claims made around resistance against evolving threats. Oversimplification and a narrow scope of evaluations remains an important issue.

**Methods And Evaluation Criteria:**

The defenses used are consistent with infectious jailbreaks in multi-agent systems. Most experiments use the Llava 1.5 7B model with CLIP RAG on toy tasks. The inspector prompt approach does not come with an analysis of its effectiveness. The adaptive adversary setting has limited evaluation.

**Other Comments Or Suggestions:**

Clarify the exact problem in the abstract and introduction, along with what the paper does. Too much of the paper is spent on discussing ‘cures’ and ‘infections’ without making the attack surface and specific method very clear.

Focus on how such attacks and defenses can appear in the real world while providing concrete directions for future research.

**Other Strengths And Weaknesses:**

Strengths
The paper presents a novel way to convert positive feedback loops in infectious jailbreak attacks into negative ones with the use of only a few agents as part of the defense. The focus on system level security is relevant and important to where the field is headed. The theory and experiments, despite their limited real-world implications, are important and the defense mechanism aligns with constraints that would be seen in real multi-agent systems.

Weaknesses
The paper’s scope is limited to the exact same underlying models interacting in a random communication pattern.
Evaluations against adaptive attacks are limited.

It is important to evaluate settings that resemble real-world use cases: i.e. agents with different underlying models and goals, structured communication networks, and adaptive attacks.

The LLM inspector used is not tested and evaluated in detail. It appears as a black-box in the paper despite its importance to the experimental results.

**Questions For Authors:**

Could the authors provide a concrete quantitative evaluation of your LLM based inspector prompt for detecting malicious outputs (along with false positive/negative rates)

Could the authors simulate a small scenario where an attacker can create multiple viruses or adaptively optimize them over several rounds?

Are there any simpler baseline level defenses that the authors can test against?

Does the effectiveness hold against agents with diverse underlying base models and RAG systems? How can the method be extended to more realistic, structured communication settings?

## update after rebuttal

I thank the authors for addressing my questions. I believe the paper has improved and have increased my score accordingly.
Please note that there seems to be some related work that should be cited:
"Multi-Agent Security Tax: Trading Off Security and Collaboration Capabilities in Multi-Agent Systems", Peigne-Lefebvre et al., https://arxiv.org/abs/2502.19145

**Relation To Broader Scientific Literature:**

The paper highlights how security measures in multi-agent literature are underexplored. It focuses on a specific aspect, infectious jailbreak attacks based on external RAG systems accessed by all agents in the system (presented in “Agent Smith: A Single Image Can Jailbreak One Million Multimodal LLM Agents Exponentially Fast”). It also relies on literature around stability analysis and simulation based evaluation for defining the problem clearly. However, the scope and impact of this setting (i.e. attack surface) remains limited in academic literature.

**Theoretical Claims:**

Proofs under the assumptions (Proposition 4.1 and 5.1) have been checked and appear valid.

---

> ### Author Rebuttal · Authors · 2025-03-31
>
> Thank you for your careful review of our paper and thoughtful comments. We hope the following responses will address your concerns.
> > ***1. About the Inspector:***
>
> We agree that the inspector is important to our method. The details of the performance of the inspector used in the paper is as below:
> ||ACC(%)|FPR(%)|FNR(%)|
> |-|-|-|-|
> |1 try|84.7|2.8|12.5|
> |3 tries|89.1|7.9|3.0|
> - Settings: The evaluation is conducted on a combination of malicious outputs from Advbench and normal(benign) outputs from the ordinary chat history of our agents. We test the inspector in 1-turn and 3-turn settings. For the multi-turn test, a sample is labeled as harmful if it is classified as harmful in ANY turn.
> - Analysis: Overall, although not able to achieve 100% accurate detection, **our inspector is effective enough for Cowpox**. This is because:
>     - **Very few benign samples will be misclassified.** There are very few Cowpox agents in the system, so less than 2% of benign samples will be misclassified.
>     - **Misclassifying benign samples has little impact on the system.** The cure crafted based on misclassified benign samples still perfectly contains its original information. So there will be almost no effects.
>     - **FNR approaches 0 when the test round increases.** The system only needs the virus to be detected in any chat round ONCE to achieve effective defense. Cowpox does not require a very strong inspector
>
> > ***2.1 Multiple Viruses***
>
> **Cowpox works well in the multiple virus scene**
> - Settings: We crafted 10 different viruses carried by random agents initially. They are based on different image and malicious outputs.
>
> - [Results](https://imgur.com/kCxmQqV).
> - Analysis: We can see there is also competition between the virus. All the infectious rate curve approaches 0 at the end of the simulation
>
> > ***2.2 Adaptively Optimized Samples Overrounds:***
>
> **Cowpox makes the system increasingly robust to similar viruses.**
> - Settings: The attacker continually collects the cure sample for the previous virus and optimizes the new virus, making it have a higher RAG score.
> - Results: Below showed the peak infected rate of viruses of each version.
>
>
> |Virus Version|0|1|2|3|
> |-|-|-|-|-|
> |Max Infected Rate|0.77|0.26|0.18|0.14|
>
> - Analysis: When the attacker continues to adapt craft the virus, it becomes harder and harder for the virus to gain a higher RAG score, according to Proposition 5.1. Therefore we can see a declining Peak infected rate.
>
> > ***3: About simple baseline***
>
> We acknowledge the importance of a good baseline. However, we think **it is difficult to compare any existing baseline with our work fairly, currently.**
> - **The infectious attack is very pioneering at present.** As the first system-level defense against such attack, there is no similar work that we can compare with.
> - **The existing defenses are only at the model level.** They only provide protection to single agents, regardless of the rest agents in the system. We must assume the defender has access to every agent to achieve reasonable performance. While cowpox works in circumstances where the access to the agents is very limited. These model-level defenses are completely ineffective in these settings.
>
> > ***4: Effectiveness in various structured communication settings.***
>
> **Cowpox is effective in various structured communication settings.**
> - Settings: We assume that the system is hierarchical. The agents are divided into 8 groups, and can only do the pair-wise chat with their group members. In each group, there are 3 manager agents, which communicate with manager agents of other groups every 4 rounds. Such hierarchical structures are often adopted in works like [AutoGen](https://arxiv.org/abs/2308.08155).
> - [Results](https://imgur.com/2gN812H).
> - Analysis: We can see that the virus infects agents more slowly, and so does the Cowpox cure the agent. This is because the hierarchical structure can temporarily isolate the infected agents.
>
> > ***5: Diverse base models and RAG systems***
>
> **The effectiveness holds when heterogeneous agents coexist in the same system**,
> - Settings: we adopt 2 VLM models (LlaVA-1.5, InstructBLIP) and 2 RAG encoders (CLIP, DINO V2) in the experiment. The agent chooses its base model and RAG encoder *randomly* initially to form a multi-agent that consists of heterogeneous agents.
> - [Results](https://imgur.com/LOQiZ0R).
> - Analysis: From the figure, The virus in this system performs worse, while Cowpox is almost equally effective. This is because the cure only targets the RAG system, therefore fewer models are involved in crafting it, making the optimization easier.
>
> > ***6: About the CLIP and BLEU score***
>
> **CLIP score and BLEU score are used to measure how well the original virus samples can be restored to normal samples by Strategy ①, rather than the system degradation.** The CLIP score and BLEU measure how far the output deviates from the benign one. A virus naturally has small scores as its outputs are manipulated.

---

### Decision · Program_Chairs · 2025-05-01

**Decision:**

Accept (poster)

**Comment:**

This paper introduces Cowpox, a defense strategy for multi-agent systems to stop adversarial jailbreaks, especially in systems that use VLMs. The main idea is to spread "cure" samples across the agents, which helps prevent infections from spreading and makes it easier for infected agents to recover. The idea is pretty cool, and the authors back it up with solid theory.

But I think there's a question of how well this would work in real-world situations. A lot of the experiments are on a specific system setup, and I wonder if it would hold up in more diverse scenarios. Some reviewers mentioned concerns about how generalizable this approach really is. That said, the authors did extend their experiments to test things like different models and communication structures, which adds to the credibility of the method. Still, the approach feels a bit narrow in scope, and there’s definitely room for more broad testing. Overall, I'm leaning towards a borderline accept, but I think it could use a bit more real-world context to really sell the idea.